# Effect of inhibition indexed by auditory P300 on transmission of visual sensory information

**Amirmahmoud Houshmand Chatroudi[1,2], Reza Rostami[2]\*, Ali Motie Nasrabadi[3], Yuko Yotsumoto[1]**

**1** Department of Life Sciences, The University of Tokyo, Tokyo, Tokyo, Japan, **2** Faculty of Psychology and Educational Sciences, University of Tehran, Tehran, Tehran, Iran, **3** Department of Biomedical Engineering, Shahed University, Tehran, Tehran, Iran

\* rrostami@ut.ac.ir

**Data Availability Statement:** Reported data and analysis scripts from all experiments in this study

## Abstract

Early electroencephalographic studies that focused on finding brain correlates of psychic events led to the discovery of the P300. Since then, the P300 has become the focus of many basic and clinical neuroscience studies. However, despite its wide applications, the underlying function of the P300 is not yet clearly understood. One line of research among the many studies that have attempted to elucidate the underlying subroutine of the P300 in the brain has suggested that the physiological function of the P300 is related to inhibition. While some intracranial, behavioral, and event-related potential studies have provided support for this theory, little is known about the inhibitory mechanism. In this study, using alpha event-related desynchronization (ERD) and effective connectivity, based on the causal (one-way directed) relationship between alpha ERD and P300 sources, we demonstrated that P300's associated inhibition is implemented at a higher information processing stage in a localized brain region. We discuss how inhibition as the primary function of the P300 is not inconsistent with 'resource allocation' and 'working memory updating' theories about its cognitive function. In light of our findings regarding the scope and information processing stage of inhibition of the P300, we reconcile the inhibitory account of the P300 with working memory updating theory. Finally, based on the compensatory behavior of alpha ERD at the time of suppression of the P300, we propose two distinct yet complementary working memory mechanisms (inhibition and desynchronizing excitation) that render target perception possible.

## Introduction

Information processing theory and improvements in signal averaging techniques ushered in a new era of electroencephalographic (EEG) studies during which the relationship between psychic events and small-amplitude potentials evoked by sensory stimuli started to surface [1]. From one of these studies focused on finding brain correlates of uncertainty came the P300, one of the early event-related potentials (ERPs) discovered in neuroscience [2]. The P300 is a positive deflection in brain waves that peaks at around 250–500 ms after onset of infrequent

are publicly available on the Open Science Framework (https://osf.io/zytfx/).

**Funding:** This research was funded by Japan Society for the Promotion of Science KAKENHI (Grant #19H01771) to YY. https://www.jsps.go.jp/english/index.html The funders had no role in study design, data collection and analysis, decision to publish, or preparation of the manuscript.

**Competing interests:** The authors have declared that no competing interests exist.

stimuli. Since its discovery, the P300 has been used widely in clinical and basic neuroscience research to assess perception and cognition in both healthy and pathological populations [3].

Despite the wide applications of the P300, attempts to elucidate its underlying mechanisms have yet to produce a cogent account of its cognitive and physiological functions in the brain. Various studies have linked the P300 to expectancy [4, 5], resource allocation and mental capacity [6, 7], categorization [8, 9] and context (working memory) updating processes [10, 11].

In addition to these studies, which have focused on cognitive mechanisms illuminating the major functional role of the P300 in the brain, there has been a line of research suggesting that P300 is an index of physiological inhibition. For example, Desmedt and Debecker [12, 13] linked the P300 to phasic prefrontal inhibition of neuromodulation of the mesencephalic reticular formation (MRF) in the telencephalon. In their study [12], the aim of which was to dissociate P300 from the positive deflection that follows the contingent negative variation (CNV), they proposed that active attention to presentation of serial stimuli prompts the MRF to maintain a definite level of DC negativity in the cortex. However, after identification of the target stimulus, the pre-frontal granular cortex inhibits the MRF, which in turn transiently decreases the negativity of the telencephalon. This event leads to generation of a P300 in the brain and marks the end of a stimulus processing epoch [13, 14].

Elbert [15], on the other hand, proposed a cortical excitability regulation model in which P300 is deemed to result from widespread suppression of extraneous neural ensembles at the time of working memory (WM) updating. According to this framework, for representations of information in the WM to be updated without interference, a large number of irrelevant neural networks must be shut off. Inhibition of these unnecessary networks at the time of the memory updating process is indexed by a widespread positive wave that is recorded as a surface P300 (see Discussion section).

Although there is no consensus as to which brain structures and physiological mechanisms give rise to the proposed inhibitory effect, a number of studies have implicated inhibition as the primary function of the P300 in the brain. For example, intracranial research has demonstrated that medial-temporal lobe P300 potentials suppress multiunit activity recorded from the hippocampal formation, amygdala, and parahippocampal gyrus [16, 17]. Moreover, behavioral studies have also pointed to an inhibitory effect by demonstrating that the process of a second stimulus in the time course corresponding to the P300's peak is impaired. For example, in a secondary task paradigm, Woodward et al. [18] demonstrated that the reaction times to probes presented at around 310–340 ms after oddball stimuli are significantly slower than those to probes presented in other time courses. Even more surprisingly, Bachanas [19] found that auditory oddball stimuli could reduce sensitivity in a subsequent visual target detection task when targets were presented around the peak latency of the auditory P300 (also see Odierna [20]). Similarly, McArthur et al. [21] found that the time course of attentional blink, which is a brief impairment of visual processing occurring 200–500 ms after a target in a rapid stream of visual stimuli, corresponds to the P300 latency and that both P300 amplitude and attentional blink react similarly to modulation of task difficulty at the group level.

Apart from physiological and behavioral studies, some EEG studies have attempted to shed light on the inhibition that is indexed by the P300 by assessing suppression of early and late evoked potentials in the brain. A study by Rockstroh et al. [22] demonstrated that preceding oddball stimuli can slow the reaction time to probes and abate probe's early evoked potentials (N1/P2) when the oddball stimuli elicited the P300. Similarly, in a study that included a dual-task paradigm consisting of merged auditory and visual oddball tasks, Nash and Fernandez [23] found that the first oddball stimuli (eliciting the auditory P300) could prevent formation of a second (visual) P300 and result in slower reaction times.

Studies that have focused on the inhibitory effect by assessing suppression of other brain potentials are important in terms of providing solid evidence for robust cortical inhibition at the time of P300's peak latency. However, these studies could not identify the stage at which the inhibitory effect is enacted or its potential consequences for perception. For example, when juxtaposing the extant findings, it is unclear whether inhibition in the early sensory processing stages (N1/P2, as reported by Rockstroh et al. [22]) or inhibition in the later cognitive processing stages (P300, as reported by Nash & Fernandez [23]), or a combination of both, slows reaction time and impairs target detection. Moreover, such studies do not yield any information about the extent of the inhibition in the cortex. Does the P300 underlying subroutine suppress a wide range of cortical areas, and thus represent a distributed general inhibitory mechanism, or does it inhibit a narrow cortical area, presumably at loci of its generation, at a specific processing stage? Finally, no study has ever explored that, in the absence of the proposed inhibition that is indexed by P300, what mechanisms will eventually contribute to target perception. Would any brain mechanism possibly compensate for the lack of inhibition? Answering these questions requires investigation of other closely related brain indices that can provide information about the processing stage and loci of inhibition as well as possible interactions in the absence of P300 inhibition.

One of these brain indices is alpha-band frequency waves. Alpha waves are 8–13-Hz oscillations that debatably originate from corticocortical (e.g., layer 5 in V2, V4 and layer 2/3 in inferotemporal cortex [24]; layer 4 in V1 [25]; higher-order visual and somatosensory areas, [26]) or thalamocortical connections [27–30]. An alpha rhythm is classically deemed to be an 'idling rhythm' [31] or an index of cortical inactivation that signals a closed thalamic gate [32]. More recent studies, have associated increases in alpha power (known as event-related synchronization, ERS) with active inhibition (e.g., attention-modulated suppression of sensory input [33–36]; pulsed-inhibition of ongoing neural activity [37, 38]; periodic inhibition of task irrelevant brain areas [39–41]). On the other hand, reduction of alpha power (known as event-related desynchronization, ERD) has been associated with an open thalamic gate [32] and is considered to be an index of cortical excitation or release of inhibition [41, 42]. Numerous studies have demonstrated a relationship between alpha ERD and attention [43], memory retrieval [44], and general arousal [45]. The functional cognitive similarities between alpha ERD and the P300 [46] encouraged researchers to investigate the nature of their relationship. By using multiple regression analysis, a significant relationship was found between the P300 and alpha ERD, such that the peak and latency of alpha ERD could be predicted by the peak and latency of P300 [47]. However, this initial finding was challenged [48] and in a recent study [49] using effective connectivity, based on Granger causality, it was recognized that there is a consistent flow of information from cortical alpha ERD sources to the sources that generate the P300 and not vice versa.

Changes in alpha power due to reflection of gating function in primary sensory areas [32, 40] and the causal (one-way directed) relationship with P300 [49] have paved a unique way for investigation of the inhibition on transmission of visual sensory information that is indexed by the P300. In this research, by adopting a dual-task paradigm [23], we presented visual oddballs in the time course that roughly corresponded to the auditory inhibition peak of the P300 (400 ms). We then hypothesized that if inhibition is widespread and implemented at stages as early as the sensory processes at thalamocortical connections, the auditory P300, at the time of incoming visual oddballs, should result in higher alpha power (event-related synchronization) in the posterior cortex. However, if the proposed inhibition acted as a suppressive function in the transmission of sensory visual information from primary sensory areas to higher areas, instead of a change in visual alpha ERD, the effective connectivity between visual alpha ERD sources and visual P300 sources would be weakened. Moreover, we hypothesized that if the

inhibition that is indexed by P300 is enacted at a specific higher information processing stage and has a locally restricted impact, the effect would be limited to suppression of the visual P300 and subsequent prolongation of reaction time. Finally, we sought to explore that, under conditions when the presumed inhibitory effect that is indexed by P300 is obviated, what possibly compensatory mechanism will eventually contribute to target perception.

## Methods and procedures

### Subjects

Twenty-four right-handed paid subjects (13 male) of mean age 23.4 (standard deviation [SD] 2.87) years studying at the University of Tehran were recruited for this research. The study was approved by the Iran University of Medical Sciences research ethics committee. All subjects provided written informed consent. All participants had normal or corrected-to-normal vision and were examined by a general physician prior to the start of the experiment. All were free of medication and none reported a significant medical history, significant head injury, psychiatric or neurological disorder, or drug or alcohol abuse.

### EEG recording

EEGs were recorded using a 64-channel DC amplifier (g.tec medical engineering GmbH, Schiedlberg, Austria) with active electrodes placed over scalp positions corresponding to the 10–10 international montage (AFz, Fp1, Fp2, AF3, AF4, F7, F3, Fz, F4, F8, FC5, FC1, FC2, FC6, T7, C3, Cz, C4, T8, CP5, CP1, CP2, CP6, P7, P3, Pz, P4, P8, PO3, PO4, O1, O2, AF7, AF8, F5, F1, F2, F6, FT7, FC3, FCz, FC4, FT8, C5, C1, C2, C6, TP7, CP3, CPz, CP4, TP8, P5, P1, P2, P6, PO7, Poz, PO8, Oz, P9, P10, and TP9). The sampling rate was set at 1,200 Hz, impedance was kept below 20 kΩ, and no online filter was applied during the recordings. The reference electrode was positioned on the right earlobe and EOG was recorded using three additional electrodes placed over the left outer canthus of the left eye, right outer canthus of the right eye, and bottom of the left eye. Stimuli were created in a MATLAB environment using Psychtoolbox-3 extension [50–52] and presented on a 15-inch LCD monitor at a viewing distance of 65 cm. Stimuli triggers were sent to the EEG amplifier using g.GAMMAbox.

### Stimuli

A dual-task paradigm consisting of merged auditory and visual oddball tasks was used [23]. In the auditory oddball task, the stimuli were 2500-Hz and 1500-Hz sinusoidal tones (80 ms in duration, 20 ms for rise/fall) with a 65-dB sound pressure level intensity, which were assigned as target (p = 0.2) and standard stimuli (p = 0.8) in a counter-balanced manner between subjects. Auditory stimuli were followed by a visual oddball task where the stimuli were $1° \times 1°$ green and red squares (duration, 80 ms) and were designated as target (p = 0.2) and standard (p = 0.8) stimuli counter-balanced between subjects. In each oddball task, the order of presentation of the stimuli was controlled by a quasi-random function so that no target stimuli of the same modality were presented in consecutive trials (presentation of target stimuli in each oddball task was separated by at least one standard stimulus). The stimulus onset asynchrony between the two oddballs was set at 400 ms and the inter-trial interval was within a random time range of 1800–2200 ms (Fig 1).

### Procedure

Each subject sat on a partially reclining chair in a dark and sound-proof room. Each block began with a 1000-ms fixation and was followed by the auditory and visual stimuli. The subject

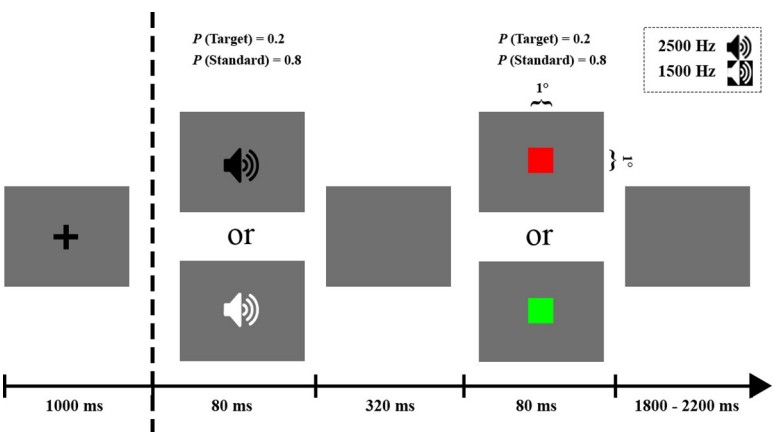

**Fig 1. Stimulus dimensions and timeline of the oddball dual-task.** Each trial started with presentation of auditory stimuli (target or standard) followed by presentation of visual stimuli (target or standard) 400 ms later. At the beginning of each block or after the auditory targets were queried, a fixation cross was displayed for 1000 ms prior to the start of the trial.

was asked to count the number of auditory targets and to press the space key as quickly as possible whenever they saw a visual target. They were instructed to pay equal attention to both modalities and keep mistakes at a minimum. Prior to the main experiment, each subject completed 40 practice trials to ensure that they understood the instructions. Once per 20 trials, a question appeared on the screen asking the subject to enter the number of auditory target stimuli they had heard (mean 4, SD 0.91, range, 1–6). Subsequently, they were given feedback to ensure that they did not ignore auditory stimuli. However, no feedback was provided for visual stimuli during the main experiment. Each subject completed 800 experimental trials divided into 4 blocks with a 7-minute rest period between blocks. After completion, for each subject, the trials were divided into the following four conditions and labeled for subsequent analysis: standard-standard (p = 0.64, n = 512), standard-target (p = 0.16, n = 128), target-standard (p = 0.16, n = 128), and target-target (0.04, n = 32).

## Analysis

Before any analysis, trials containing incorrect behavioral responses to visual stimuli (commission or omission errors) were removed (0.3% of trials). Moreover, if a subject failed to report the correct number of auditory targets (with a tolerance of ±1 difference), the preceding 20 trials were removed (0.4% of trials). This measure was adopted to ensure that the data (both behavioral and EEG) reflected task-related cognitive processes and contained trials in which equal attention was paid to both modalities.

**Behavioral data.** To assess the inhibitory effect on behavioral responses that is indexed by P300, reaction times to visual stimuli under the standard-target and target-target conditions were analyzed. For each participant, standard-target and target-target reaction times were averaged across conditions and examined using the Shapiro-Wilk normality test [53]. After ensuring that the normality assumption was not violated [W(24) = 0.92, $p$ = 0.09], these averages were subjected to a pairwise $t$-test (one-tailed) to test for any significant increase in the target-target condition relative to the standard-target condition.

**Electroencephalography data.** *Preprocessing.* The EEG data were first down-sampled to 600 Hz. Subsequently, a low-pass filter at 30 Hz and a high-pass filter at 0.5 Hz were applied. Continuous data were segmented into 2400-ms epochs with an 800-ms pre-stimulus period

time-locked to the onset of the auditory stimuli. The epochs were inspected for major contamination, and noisy channels were removed and interpolated. After visual inspection for major contamination, independent component analysis was performed to remove EOG, movement, and muscle artifacts. At this stage, the EEG data from one subject were excluded from further analysis due to excessive muscle contamination.

*Event-related potentials*. Epochs were baseline-corrected using the pre-stimulus period. Average waveforms were computed for each subject under each condition. Auditory and visual P300s were, respectively, defined as the highest peak around 250–400 ms and 650–800 ms after onset of the auditory stimulus in the Pz electrode. To obtain a more stable measure of the P300 [54], the amplitude of the P300 was computed using the average amplitude in time points corresponding to ±1 SD of the grand peak latency for each condition. The baseline-to-peak amplitudes of P300 at the group level were subsequently compared separately for each modality between the four conditions using repeated-measures analysis of variance (rANOVA). The sphericity assumption was assessed using the Mauchly test; when violated, Greenhouse-Geisser correction was applied. Bonferroni correction was used for the multiple comparison problem.

Moreover, in order to explore significant differences in amplitude across all time points of the ERP waves under different conditions, a one-to-one paired *t*-test comparison between ERPs under the four conditions was performed using the pop_comperp() function implemented in EEGLAB [55]. The false discovery rate method was implemented using the fdr() function of EEGLAB to correct the multiple comparison problem. Finally, areas that were significant ($p < 0.001$) were highlighted.

In addition to amplitude, in order to assess the possible effects of the experimental conditions on P300 latency values, auditory P300 latency was compared between the target-standard and target-target conditions and visual P300 latency between the standard-target and target-target conditions. Latency was defined as the time point corresponding to the peak of the P300 in each modality. After ensuring that there was no breach of the normality assumption using the Shapiro-Wilk test [$W_{Auditory}(23) = 0.91$, $p = 0.06$; $W_{Visual}(23) = 0.94$, $p = 0.25$], visual and auditory P300 latencies were compared under the aforementioned conditions using a pairwise *t*-test (two-tailed).

*Alpha event-related desynchronization*. In order to assess how the inhibitory effect that is indexed by the auditory P300 might influence the processing of visual inputs in early sensory areas, visual alpha ERD was compared between the standard-target and target-target conditions in four stages for hypothesis-driven analysis and one additional stage for exploratory analysis.

1. ***Computation of time-frequency distribution***: to reveal both phase-locked and non-phase-locked dynamics, power changes were calculated for each condition using complex Morlet wavelet decomposition on single-trial EEG data with custom MATLAB scripts according to the following formula [56]:

$$CMW = e^{-t^2/2s^2} e^{i2\pi ft}$$

The length of the Morlet wavelet (t) was set to 4 s (range +2 to -2 centering on 0 s based on recommendations in [56]). The central frequency parameter (f) was adjusted to the range of 1–30 Hz in 0.5-Hz increments. The standard deviation or the width of the Gaussian (s) was obtained using the following formula:

$$s = \frac{n}{2\pi f}$$

The number of wavelet cycles (n) was set to the logarithmically spaced range of 4–10 cycles so that it increased proportionately with central frequency (thus, the range of standard deviation (s) was 0.63–0.05). An increase in the number of cycles as a function of central frequency was implemented to balance the precision of time and frequency [56].

2. ***Baseline correction***: the computed time-frequency distributions were corrected using the -400 ms to -100 ms baseline period by implementing decibel normalization according to the following formula [56]:

$$ER(t,f) = 10 \log_{10} F(t,f)/R(f)$$

In this formula, *F(t, f)* is the power of the signal at any given time point (t) and frequency (f). Furthermore, *R(f)* is the power of frequency (f), which is averaged in the baseline period.

3. ***Delineating the time-frequency region of interest***: The time-frequency region of interest (ROI) spanned the 8–13-Hz frequencies, corresponding to the alpha band limits, and a time interval of 616–822 ms. Similar to the study by Peng et al. [49], the lower limit of this interval is 100 ms before the grand peak latency of the visual P300 (716 ms for the standard-target and target-target conditions combined). However, the upper limit is selected such that it is 2 SDs away from the mean reaction time for the visual target under the standard-target condition. The upper limit is chosen conservatively to eliminate possible muscle-related alpha perturbations.

4. ***Statistical testing***: The alpha ERD in the time-frequency ROI was extracted and averaged for the PO3, PO4, PO3, PO4, PO7, PO8, O1, O2, and Oz electrodes (again, the choice of electrodes was made very conservatively in order to minimize the chances of pre-movement alpha modulations degrading the inference) under the standard-target and target-target conditions. Subsequently, after ensuring that the normality assumption was not violated using the Shapiro-Wilk test [$W(23) = 0.95$, $p = 0.36$], the averaged alpha ERD was compared between the two conditions using a pairwise *t*-test (two-tailed).

5. ***Permutation-based statistical testing***: In order to explore significant power changes in all frequencies and time points between the standard-target and target-target conditions, time-frequency distributions for both conditions were extracted from the same electrodes as in stage 4. Subsequently, for each subject, the label of subject-level averaged standard-target and target-target conditions were shuffled; the grand average difference between the two conditions was then computed and stored. This process was repeated 10,000 times to generate the null distribution of no group-level difference between the two conditions. Statistical significance ($p < 0.01$, two tailed) was eventually assessed under the null hypothesis using the pixel-based multiple comparison correction method [56]. In this method, each time-frequency point (pixel) in the grand average difference between two conditions is compared against the 99.5 percentile of the maximum value and 0.5 percentile of the minimum value of iterations so that any time-frequency pixel with a value higher or lower than the aforementioned percentiles is regarded as a significant power dynamic (for detailed explanation of this method see [56]). In comparison to cluster-based correction (see effective connectivity subsection), pixel-based multiple comparison correction, is considered to be a more stringent correction method for time-frequency maps [56]. Since the exploratory analysis was performed post hoc to the hypothesis-driven analysis, we used this method together with a strict p-value ($p < 0.01$) to assure our results have sufficient statistical rigor.

*Source localization*. For source localization of the visual P300 and visual alpha ERD as the base condition, only data from the standard-target condition were analyzed.

*Visual P300*. The location of the visual P300 sources was estimated using LORETA-KEY alpha software [57]. LORETA (low-resolution brain electromagnetic tomography) computes three-dimensional linear solutions for the EEG inverse problem within a three-shell spherical model of the head that includes the scalp, skull, and brain compartments and is widely used for source localization of EEG data. The location and strengths of the regional sources for subject-level ERPs in the time range of ±24-ms intervals around the visual P300 peak time range (±1 SD of grand latency in the standard-target condition) were computed. The resulting sLORETA matrices were then analyzed using statistical nonparametric mapping, which is a statistical toolbox [58] used for voxel-level non-parametric permutation tests in functional neuroimaging experiments. Using the general linear model, statistical nonparametric mapping constructs pseudo *t*-statistic images, which are then assessed for significance using a standard non-parametric multiple comparisons procedure. Here, we used a single-group *t*-test statistic and 5000 randomizations for each condition separately to identify voxels with significant activation (p < 0.01).

*Visual alpha ERD*. The cortical sources of visual alpha ERD were also estimated using the LORETA-KEY alpha software. For the baseline and visual stimuli periods, brain waves in the time range of -400 ms to -100 ms and 616 ms to 822 ms were extracted from each epoch and concatenated to form pseudo-EEG data (separately for the baseline period and the visual stimuli period). The pseudo-EEG data for each period were then transformed to the frequency domain using the cross-spectrum function in LORETA and restricted to alpha-range oscillations (8–13 Hz). Next, the baseline and visual stimuli cross-spectrum activation voxels were compared using paired-sample *t*-tests. Statistical nonparametric mapping with 5000 randomizations was also used to assess the statistical significance of the results (p < 0.001).

*Effective connectivity*. The effective connectivity between the standard-target and target-target conditions was compared to determine if the inhibition that is indexed by the auditory P300 affects flow of information from visual alpha ERD sources to visual P300 sources and vice versa. This comparison consisted of the following three steps.

1. **Extracting brain waves**: Single-trial brain waves from visual P300 and visual alpha ERD sources were extracted using Brainstorm [59]. To do this, voxels that were marked as sources of P300 by LORETA were approximated to the nearest corresponding voxels in the Cortex_15002V template of Brainstorm and demarked as ROI-1. Given that areas specified by LORETA as sources of visual alpha ERD included a diverse range of voxels, with some having very low values (see Results section), the first 10 percent of voxels with higher activation were approximated to the Brainstorm cortex template and marked as ROI-2. After delimiting the ROIs, the forward problem was solved using a three-shell spherical model of the head for each subject. The noise covariance was then computed using a baseline interval of -800 to 0 ms. Finally, using the minimum norm imaging method, the relative value of the current density maps for each ROI was extracted using the median eigenvalue of noise covariance regularization.

2. **Computation of Granger causality**: The Granger causality was computed on extracted brain waves using the Source Information Flow Toolbox (SIFT) [60, 61]. First, the data for each condition were down-sampled to 125 Hz, and to eliminate possible edge effects, 200 ms from the beginning of each epoch and 400 ms from the end of each epoch was discarded (thus, the length of the epochs was reduced to -600 ms to 1400 ms). Next, in order to enhance the fitness of the model and equalize the model orders between conditions, the number of standard-target epochs was made equal to the number of target-target trials. To

this end, first, the standard-target trials (n = 128) were linearly segmented by the number of target-target trials (n = 32). Next, one out of the total trials in each segment (n = 4) was randomly selected. By equalizing the number of epochs using this method, random trials were selected from all time segments in the experiment. After equalization of trial numbers, temporal and ensemble normalization were applied to the data (removing the average and dividing by standard deviation). Subsequently, for each subject, the model order was selected and fitted using the information criteria provided in SIFT. SIFT implements time-varying (adaptive) multivariate autoregression modeling by using sliding-window adaptive vector autoregression (VAR). One of these sliding-window-based autoregressive models is Vieira-Morf, which yields better coefficient estimates for small sample sizes than other models [61]. Hence, in this study, we used the Vieira-Morf model with a 500-ms window length and a 10-ms step size. This window length covers approximately one cycle of 2-Hz oscillations and thus can model the flow of information from 2 Hz upwards. After model fitting, the models were validated for each subject using the whiteness of residuals, percent consistency, and model stability (Table 1). The effective connectivity between sources of visual P300 and visual alpha ERD was explored between 2 Hz to 30 Hz in 0.5-Hz increments using a direct-directed transfer function (dDTF08), which is a multivariate Granger-based method that distinguishes between direct and indirect flows and yields reliable patterns of connection [62]. After fitting separate vector autoregressive (VAR) models to a sequence of highly-overlapping time windows, by Fourier transforming the fitted time-varying VAR coefficients, SIFT extracts complex spectral density [61]. The spectral power density is then reported in dB units, 10*log10 of complex spectral density.

3. **Permutation-based statistical testing**: The permutation testing was performed separately for each direction of flow of information (from visual alpha ERD sources to visual P300 sources and vice versa). In order to specify significant portions of effective connectivity time-frequency patterns, the null distribution of no difference between the two conditions was created by shuffling the standard-target and target-target labels for subjects and subtracting the new sets from each other over a course of 10,000 iterations. In the next step, the grand difference between the two conditions was normalized by subtracting it from the average and dividing it by the SD of the null distribution. Next, the time-frequency points that were significant at $p < 0.05$ (two-tailed) were demarcated. Cluster-based correction was then performed for the multiple comparison problem [56]. When using this method, the distribution of the largest adjacent points that form significant time-frequency clusters at $p < 0.05$ (two-tailed; precluster threshold) under the null hypothesis are extracted from the permuted data. After obtaining this distribution, the cluster size value, which corresponds to the 95th percentile of the distribution, is acquired (cluster correction threshold; $p < 0.05$). Subsequently, all clusters observed in normalized grand differences between

**Table 1. Validation parameters of fitted models based on percent residual whiteness, percent consistency, and percent stability indices under standard-target and target-target conditions for 23 subjects.**

| Criterion | Sub-criterion | Mean | SD |
|---|---|---|---|
| **Percent residual whiteness** | Ljung–Box | 98.27 | 2.27 |
| | Autocorrelation function | 88.51 | 8.4 |
| | Box–Pierce | 98.58 | 2.09 |
| | Li–McLeod | 98.08 | 2.4 |
| **Percent consistency** | | 86.72 | 1.17 |
| **Percent stability** | | 100 | 0 |

conditions that are equal or larger than that value are marked as significant portions of the data (for detailed explanation of this method see [56]). In comparison to pixel-based correction, cluster-based method is a less stringent multiple comparison correction. We used this method together with a less strict p-value (p < 0.05) to allow for detection of more areas of difference between two conditions across whole frequency and time point space separately for flow of information from visual alpha ERD sources to visual P300 sources and vice versa.

## Results

### Behavioral data

Reaction times for visual targets had longer latencies under the target-target condition (mean 619 ms, SD 72.88) than under the standard-target condition (mean 590 ms, SD 65.54). A pairwise *t*-test showed a significant difference [t(23) = -2.707, p = 0.006] in reaction time between the two conditions, indicating that the reaction times for visual targets were significantly delayed when preceded by auditory targets than when they were preceded by standard auditory stimuli.

### EEG

**Amplitude of the P300.**  The Mauchly's test result indicated that the sphericity assumption was violated [$\chi^2(5) = 21.26$, p = 0.001]. Therefore, the degrees of freedom were corrected using Greenhouse-Geisser estimates of sphericity ($\varepsilon = 0.62$). The rANOVA performed on the auditory P300 amplitude under each of the four experimental conditions showed a significant effect for condition [F(1.87, 41.32) = 54.79, p < 0.001]. Post hoc comparisons using *t*-tests with Bonferroni correction revealed that the auditory P300 under the target-standard condition (mean 4.23, SD 3.90) and target-target condition (mean 4.24, SD 3.96) were significantly different from the standard-standard (mean -0.79, SD 2.20) and standard-target (mean -1.70, SD 1.80) conditions (p < 0.001 for both comparisons). Moreover, they showed no significant difference in auditory P300 amplitude between the standard-standard and standard-target conditions or between the target standard and target-target conditions (p = 0.058 and p = 1.00, respectively). Taken together, these results show that our auditory target successfully elicited the P300 in comparison with standard stimuli and that the amplitude of this P300 was not different between target-target and target-standard conditions.

For the visual P300, Mauchly's test indicated that the assumption of sphericity was violated [$\chi^2(5) = 26.26$, p < 0.001]. Hence, Greenhouse-Geisser correction was applied ($\varepsilon = 0.57$). A significant effect for condition was detected by rANOVA computed on visual P300 amplitudes [F(1.7,37.59) = 46.73, p < 0.001]. Post hoc comparisons with Bonferroni-adjusted p-values indicated that the visual P300 amplitude under the standard-target condition (mean 13.00, SD 6.42) was significantly larger than under the target-target (mean 5.85, SD 5.57), target-standard (mean 5.17, SD 3.90), and standard-standard (mean 4.16, SD 2.56) conditions (p < 0.001 for all third comparisons). Moreover, they showed that there were no significant differences in P300 amplitude between the target-target condition and the target-standard or standard-standard condition (p = 1.00 and p = 0.656, respectively). Altogether, these results showed that visual target stimuli, in comparison with visual standard stimuli, were capable of eliciting the P300, as evident in comparison of the standard-target condition with the target-standard and standard-standard conditions. Moreover, it was shown that the visual P300 under the target-target condition was suppressed (as evident in the comparison of the target-target condition with the standard-target condition). Furthermore, the suppression occurred such that there

was no difference between the amplitude of the visual P300 under the target-target condition and that under the target-standard condition or standard-standard condition.

Exploratory one-to-one comparisons of grand average amplitude differences between conditions also conformed to the general pattern obtained using rANOVA, as shown in Fig 2. However, in addition to suppression of the visual P300 amplitude (Fig 2C–2E), this analysis also pointed to a significant increase in the visual P100 amplitude. This amplitude enhancement was detected for both visual standard and target stimuli whenever they were preceded by auditory targets and thus were presented at around the peak of the auditory P300 (Fig 2A, 2C, 2E and 2F).

**Latency of the P300.** There was no significant difference in auditory P300 latency between the target-target (mean 362, SD 25.39) and target-standard (mean 360, SD 22.10) conditions [t(22) = 0.43, p = 0.67]. Similarly, comparison of the visual P300 latency under the target-target (mean 710, SD 29) and standard-target (mean 719, SD 24) conditions did not reveal any significant difference [t(22) = 1.45, p = 0.16].

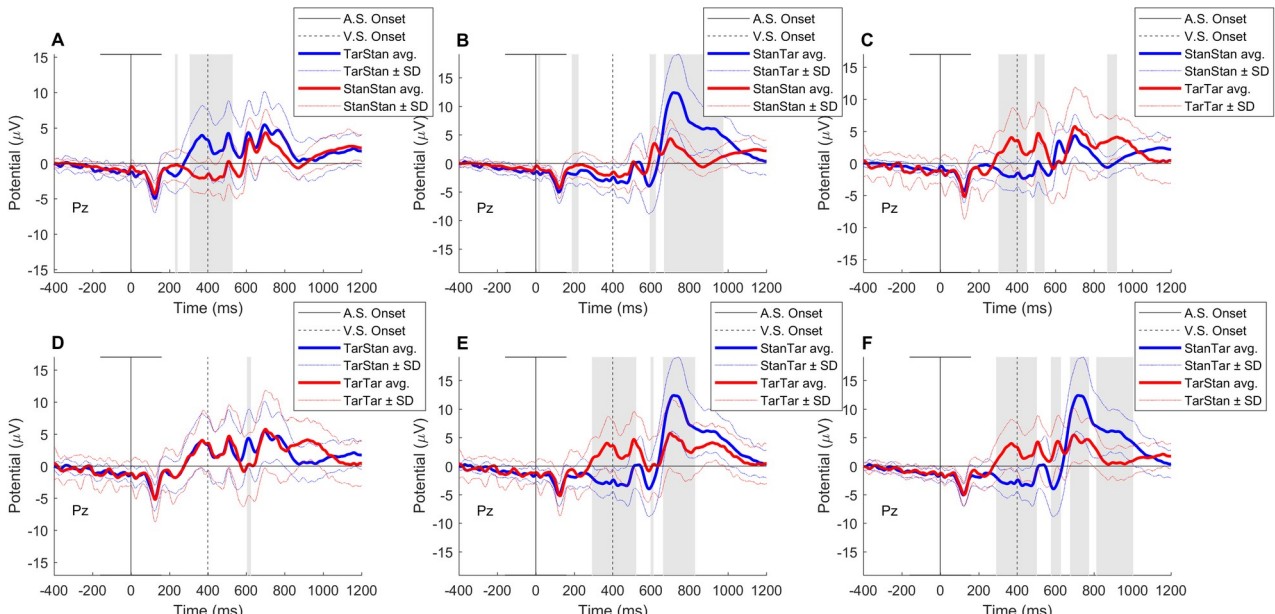

**Fig 2. Exploratory one-to-one comparison of grand average event-related potential waves between conditions using the paired *t*-test with correction for the false discovery rate.** The highlighted areas correspond to an amplitude difference with a p-value < 0.001. The solid black line indicates the onset of the auditory stimuli (AS) and the black dashed line shows the onset of the visual stimuli (VS). (A) Comparison of standard-standard and target-standard conditions revealed a significant difference between amplitudes in the time range of 308–528 ms containing two peaks. The first peak corresponds to the auditory P300 and the second peak to the visual P100. (B) Comparison of standard-standard and standard-target conditions showed a major significant difference between amplitudes in the time range of 670–975 ms, which corresponds to the time course of the visual P300. (C) Comparison of the standard-standard and target-target conditions revealed major significant differences in amplitude in the range of 303–450 ms, corresponding to the auditory P300 and in the range of 493–540 ms, which corresponds to the visual P100. However, no major difference in amplitude was detected in the time range of the visual P300, which suggests that the visual P300 has been suppressed. (D) Comparison of the target-standard and target-target conditions did not show any major differences in amplitude between the two conditions. This includes the time range of the visual P300, suggesting that the visual P300 has been suppressed under the target-target condition. (E) Comparison of the standard-target and target-target conditions indicates significant differences in amplitude in the range of 295–520 ms with two peaks corresponding to the auditory P300 and the visual P100, respectively. Moreover, the differences in amplitude in the time range of 665–828 ms, which corresponds to the time course of the visual P300, were flagged as significant, indicating that the visual P300 amplitude was suppressed under the target-target condition. (F) Comparison of the standard-target and target-standard conditions revealed significant differences in amplitude in the time range of 291–501 ms with two peaks corresponding to the auditory P300 and the visual P100. Moreover, differences in amplitude in the approximate time range of 676–1002 ms, corresponding to the visual P300, were significant. Panels C, D, and, in particular, panel E indicate suppression of the visual P300 following the auditory P300. Panels A, C, E, and F indicate enhancement of the amplitude of the visual P100 following the auditory P300. This figure also confirms that the visual stimuli had been presented (onset of VS) within the range of the peak auditory P300 where the presumed inhibition that is indexed by P300 was at its highest.

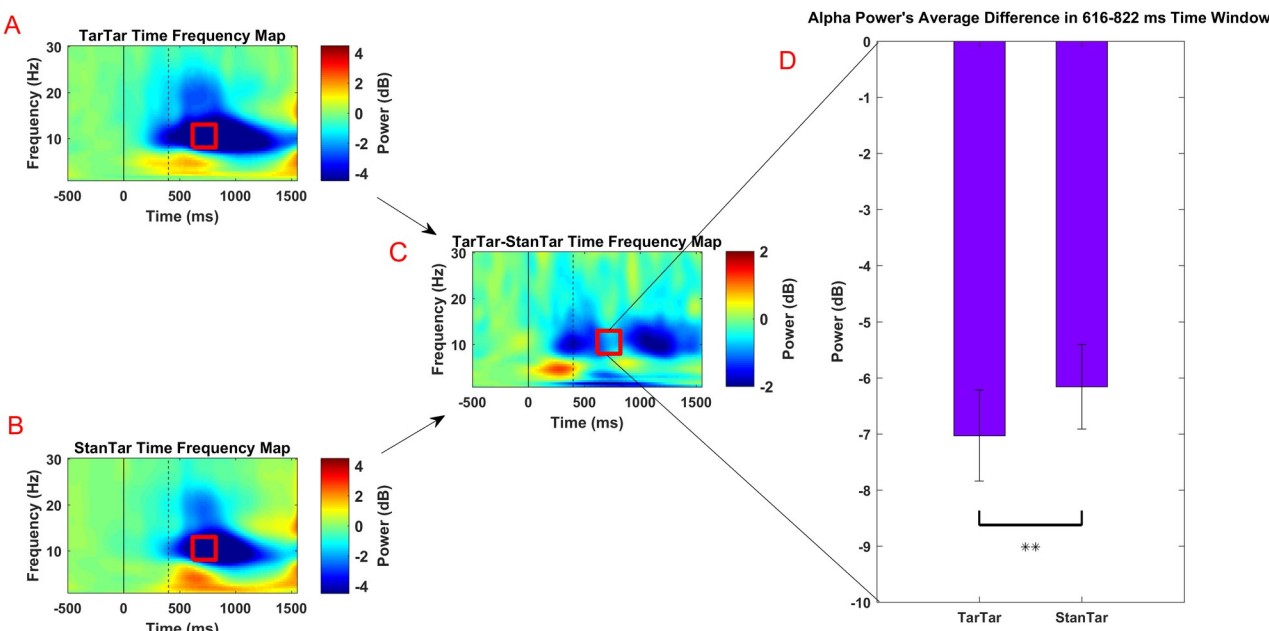

**Fig 3. Comparison of time-frequency map of power difference between target-target and standard-target conditions.** (A) Time-frequency map under the target-target condition normalized with the pre-stimulus baseline showing major alpha desynchronization after presentation of the auditory target (230 ms) up to the end of the epoch interval. (B) Time-frequency map under the standard-target condition normalized with the pre-stimulus baseline showing major alpha desynchronization around the time of onset of visual stimuli (400 ms) up to the end of the epoch interval. (C) Time-frequency difference map under the target-target and standard-target conditions. (D) Comparison of power in the time-frequency ROI between the target-target and standard-target conditions. The red square indicates the ROI where the effect of the inhibition that is indexed by the auditory P300 was expected to be observed on the time-frequency map of difference between target-target and standard-target conditions (8–13 Hz, 616–822 ms; hypothesis driven analysis, see Method section). This ROI is also marked on panels A and B with red square to indicate its relative position separately on baseline-corrected target-target and standard-target time-frequency maps. Two asterisks indicate statistical significance at p < 0.01. The solid black line indicates the onset of auditory stimuli and the black dashed line shows the onset of visual stimuli.

**Alpha ERD.** Comparison of the alpha ERD in the time-frequency ROI between the target-target (mean -7.02 dB, SD 3.89) and standard-target (mean -6.16 dB, SD 3.60) conditions unexpectedly revealed a significant reduction in alpha power [t(22) = -3.51, p = 0.002] (Fig 3C and 3D). The desynchronization of alpha at this time interval reflects over-activation of the occipital and occipitoparietal areas in response to visual target stimuli under the target-target condition in comparison with the standard-target condition. Exploratory permutation-based power-perturbation analysis (Fig 4) indicated significant alpha (8.5–11.5 Hz) power desynchronization in the time range of 298–516 ms. Reduction of alpha in this time interval under the target-target condition relative to the standard-target condition was expected due to the process of auditory target stimuli [63, 64]. However, similar to the parametric t-test results, albeit with differences in exact time interval due to the higher sensitivity of exploratory analysis, alpha (8–13.5 Hz) in the time range of 926–1303 ms was significantly desynchronized. This analysis also indicated a reduction of delta (centering at 1.5 Hz) power in the time range of 453–965 ms. Overall, analysis of the ERD did not reveal any inhibitory effect of the auditory P300 on visual alpha power; rather, it pointed to an over-desynchronization of visual alpha under the target-target condition.

**Source localization.** *Visual P300.* Source localization using the LORETA solution identified Brodmann areas 7 and 31, including the precuneus, posterior cingulate cortex, and cingulate cortex (Fig 5), as primary sources of the visual P300 (p < 0.01), with highest activation in the precuneus (Montreal Neurological Institute coordinates [–15 –60 25]).

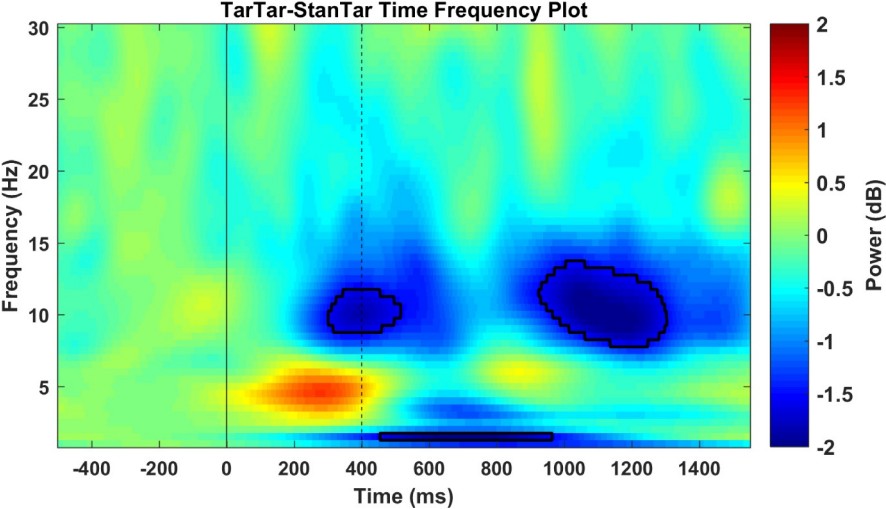

**Fig 4. Exploratory analysis of the time-frequency map of power difference between the target-target and standard target conditions (Fig 3C).** The demarcated areas correspond to time-frequency differences with a p-value < 0.01 using permutation-based statistical testing with pixel-based multiple comparison correction. The first highlighted area (on the left) indicates desynchronization of alpha in response to auditory target stimuli. The second highlighted area (on the right) indicates over-desynchronization of alpha in response to a visual target under the target-target condition. The third highlighted area (on the bottom) corresponds to desynchronization of the delta band frequency. The solid black line indicates the onset of auditory stimuli and the black dashed line shows the onset of visual stimuli.

*Visual alpha ERD*. For visual alpha ERD sources, the LORETA solution specified a relatively large number of voxels, with the highest values being located mainly in Brodmann areas 18 and 19, including the medial occipital gyrus and cuneus (p < 0.001; Fig 6). Moreover, some scattered voxels from other areas in the cortex (such as the inferior frontal gyrus (Montreal Neurological Institute coordinates [−50 45 −10]) were also marked as significant visual alpha ERD sources. Given that these voxels were relatively dispersed and had lower activation values, only the first 10 percent of voxels with the highest values (excluding voxels marked as

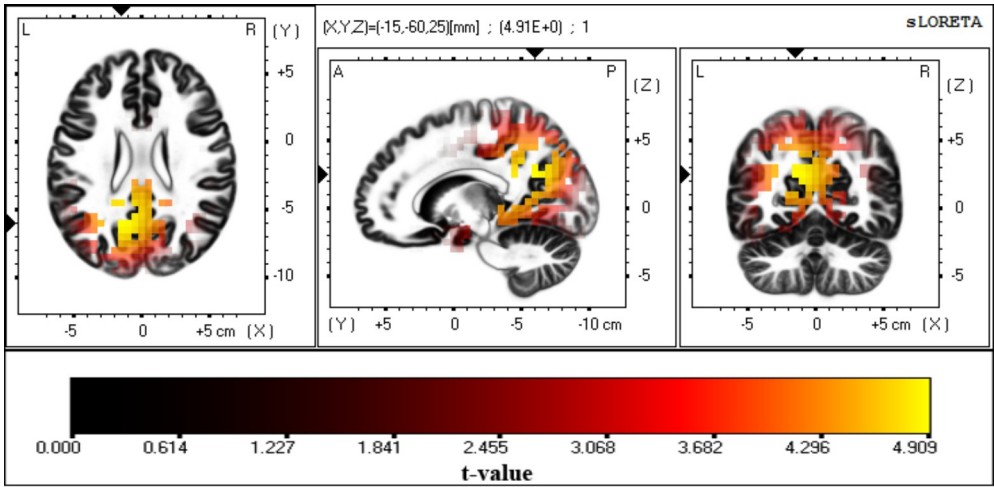

**Fig 5. Source localization of the P300.** Images show the anatomical estimation of unthresholded P300 sources from the LORETA solution using a single group *t*-test with 5000 randomizations. Maximum significance was observed over the precuneus and posterior cingulate cortex (*t*-value > 4.47 = p < 0.01).

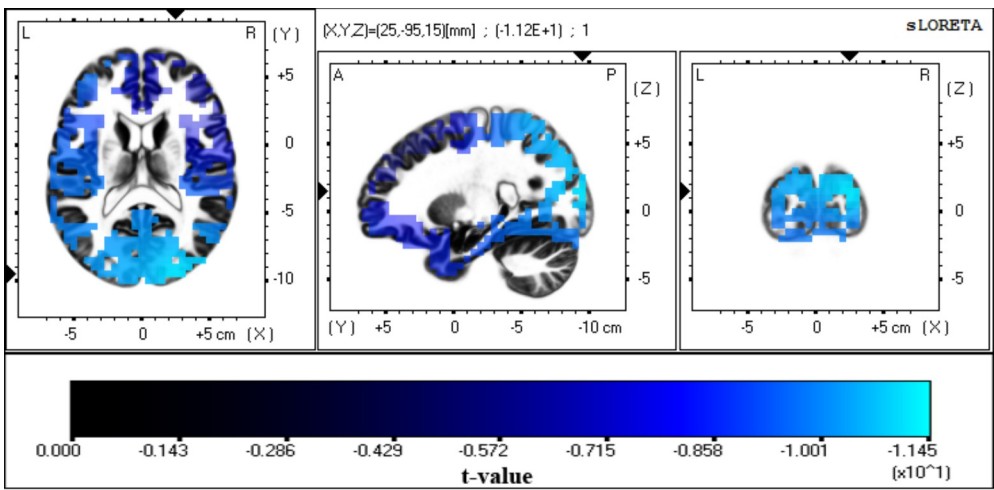

**Fig 6. Source localization of visual alpha ERD.** Images showing the anatomical estimation of unthresholded visual alpha ERD sources from the LORETA solution using paired *t*-tests with 5000 randomizations. A relatively large number of voxels were marked as significant, with the highest values being observed over the medial occipital gyrus and cuneus (*t*-value < -5.77 = p < 0.001). Only the first 10 percent of voxels with the highest values were selected as an ROI for effective connectivity analysis (see text). ERD, event-related desynchronization.

significant from the frontal lobe) were specified as a visual alpha ERD ROI for subsequent effective connectivity analysis. The ensuing demarcated ROIs (marked in red in the upper panels of Fig 7) are consistent with previous studies reporting visual alpha ERD sources of interest in the occipital and occipitoparietal areas [49, 65, 66].

**Effective connectivity.** An effective connectivity analysis was performed to assess if the inhibition that is indexed by P300 affects the flow of information between visual alpha ERD and visual P300 sources. Permutation-based statistical analysis of time-frequency distributions resulting from effective connectivity computations showed no significant difference in the flow of information from visual alpha ERD sources to visual P300 sources between the target-target and standard-target conditions (Fig 7, left column). However, this analysis revealed a significant difference (p < 0.05) in the flow of information from visual P300 sources to visual alpha ERD sources with regard to alpha band frequency, which expanded from 8.5–9.5 Hz at 280 ms after the onset of auditory stimuli to 8–15.5 Hz at 944 ms and was sustained up to the 1104-ms time point (Fig 7, right column). Taken together, effective connectivity analysis did not indicate any inhibitory effect that is associated with the auditory P300 on the flow of information from visual alpha ERD sources to visual P300 sources. However, in reverse, there was a major power reduction in the flow of alpha band frequencies from visual P300 sources to visual alpha ERD sources under the target-target condition.

## Discussion

In the present study, we investigated the characteristics of the inhibitory effect that is indexed by the auditory P300 on formation of the visual P300 [23]. We traced the locus of the inhibition with regard to stages of visual processing based on the gating function of alpha waves [32, 40] in sensory areas as well as the causal (one-way directed) flow of information from visual alpha ERD sources to visual P300 sources [49]. It is important to note that any inhibitory effect associated with P300 is in fact driven by a subroutine that is summoned based on task demands. Thus, in this study, the inhibitory effect that is indexed by the auditory P300 is

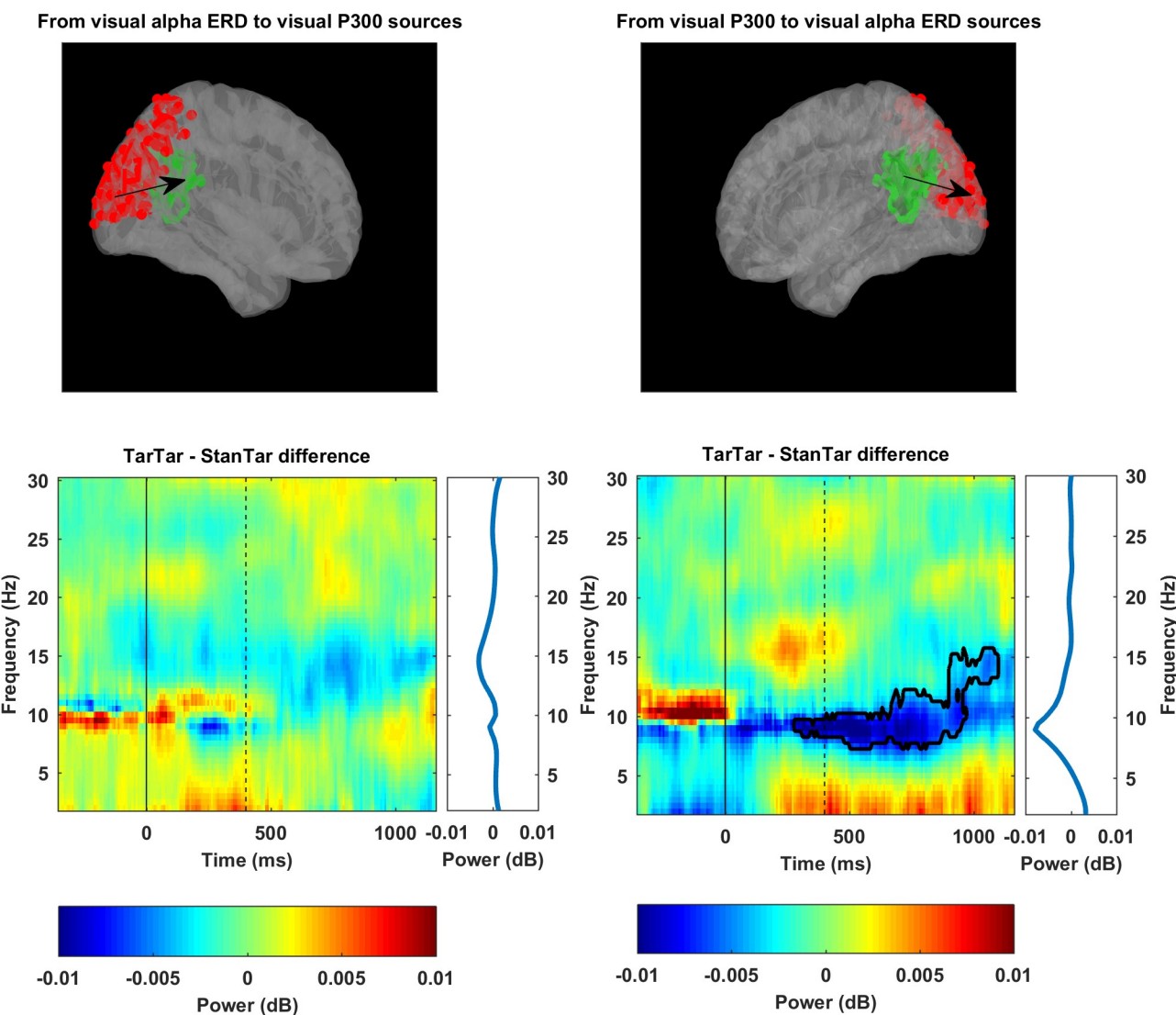

**Fig 7. Regions of interest and time-frequency maps of effective connectivity power difference between target-target and standard-target conditions.** The upper row shows left and right views of the same cortex template containing the ROIs from which brain waves were extracted and effective connectivity was computed. ROI marked by green refers to visual P300 sources. ROI marked in red refers to the first 10 percent of voxels with the highest activation values specified as visual alpha ERD sources. The left time-frequency map on the bottom row indicates no significant spectral power density difference in the flow of information from visual alpha ERD sources to visual P300 sources across conditions. However, the highlighted time-frequency area in the right image on the bottom row indicates significant differences in the flow of alpha power reduction from visual P300 sources to visual alpha ERD sources ($p < 0.05$). The subplots on the right side of time-frequency maps show spectral power density differences between two conditions (corresponding to the time-frequency maps on their left) that are averaged across the whole epoch (starting from time = 0 ms up to the end of the epoch). The spectral power density subplot of flow of information from visual P300 to visual alpha ERD sources reveals a major power reduction difference between two conditions that peaks at 9 Hz. The solid black line indicates the onset of auditory stimuli and the black dashed line shows the onset of visual stimuli. ERD, event-related desynchronization; ROI, region of interest; dB, unit of spectral power density (see Method section).

exerted by the underlying mechanisms that are required for oddball target detection and/or the updating of oddball counts in WM.

Using a dual-task paradigm, two essential conditions were created: (1) a standard-target condition whereby visual targets were preceded by standard auditory stimuli and thus were presented at the time when presumably no inhibition was in action and (2) a target-target condition whereby visual target stimuli were presented at around the time when the inhibition

enacted by the auditory P300 was at its maximum. Comparison of these two conditions with regard to behavioral data demonstrated that the reaction time for visual targets, when preceded by auditory targets, was significantly delayed. This outcome, in accordance with previous studies [18, 67], is deemed to be a behavioral indicator of the inhibition that is associated with P300. Moreover, this finding was supported by comparisons of P300 amplitude, which showed that the visual P300, when preceded by the auditory P300, was suppressed such that there was no difference between the visual target and standard stimuli in terms of evoked amplitude. While this is in agreement with the finding by Nash and Fernandez [23] that P300 can inhibit formation of a late evoked potential such as the P300, our exploratory ERP analysis showed that when following auditory targets, the P100 evoked in response to standard and target visual stimuli is not inhibited; rather, it is encouraged. However, this amplitude enhancement should be interpreted with caution as the visual P100 is added on the top of the preceding auditory P300. Thus, we conservatively interpret this enhancement as a mere indication that the inhibition that is indexed by P300 has not suppressed the formation of the subsequent visual P100.

This finding seems to be at odds with the study by Rockstroh et al. [22] where early evoked potentials (peak-to-peak amplitude of the N1/P2 of the probe) were suppressed following odd-ball target stimuli. However, one possible explanation for this conflict can be put forward based on the distinct functions that P100 and P200 (P2) reflect in the brain. P100 originates from the occipital cortex, and its amplitude is sensitive to exogenous low-level features such as luminance, contrast, color, and spatial frequencies [68], as well as endogenous top-down variables such as selective attention [54, 69]. The P100 is believed to result from feedforward flow of activity in sensory pathways [70]. The P200, on the other hand, has possible generators in the frontal lobe, its amplitude is larger for infrequent target stimuli [54], and it is deemed to be an index of context updating in the prefrontal cortex [71]. Consequently, our finding is different from that of Rockstroh et al. [22] in that it shows when visual stimuli are presented at around the peak of a preceding auditory P300, up to a specific sensory stage in the occipital cortex, stimulus processing is intact (which is evident by amplitude of early evoked potentials such as P100). Nevertheless, this intactness is followed by disfacilitation in upcoming processing stages, which possibly contain context updating in anterior regions (P200; as pointed out above) or WM processes in parieto-central areas (P300; as will be discussed later in this section).

Although ERP analysis of early components provided valuable insight, in our design, where stimulus onset asynchrony between auditory and visual stimuli is relatively short, there is an increased chance of late auditory and early visual components overlapping each other, thereby increasing the likelihood of true interactions being smeared. Hence, in order to pinpoint the loci of inhibition in the stream of afferent information, we tracked the processing of visual sensory signals in the primary visual cortex based on changes in alpha ERD as an index of the thalamic gateway [30, 32].

An alpha power comparison between target-target and standard-target conditions did not indicate any event-related synchronization in the occipital and parieto-occipital regions during the time window that corresponded to suppression of the visual P300. Surprisingly, the power of alpha in that time window was decreased. Because alpha event-related synchronization is considered to indicate inhibition [33–41], its absence in our study suggests that the P300 underlying subroutine does not suppress the flow of information from thalamocortical connections to primary visual areas. However, in a reverse pattern, our finding of over-desynchronization of alpha ERD signals that the primary visual cortex is over-activated when the visual P300 is inhibited by a preceding auditory P300. Nevertheless, due to the volume conduction problem in EEG methodology [72], electrodes placed over posterior regions can also capture alpha ERD in networks responsible for auditory processing. Therefore, the over-

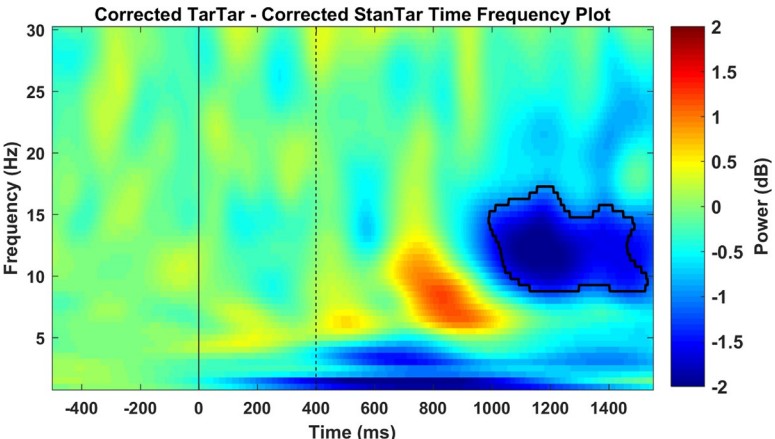

**Fig 8. Time-frequency map of the power difference between corrected target-target and corrected standard target conditions.** The demarcated area corresponds to time-frequency difference points with a p-value < 0.01 using permutation-based statistical testing with cluster-based multiple comparison correction. The solid black line indicates the onset of auditory stimuli and the black dashed line shows the onset of visual stimuli. A corrected target-target time-frequency plot was obtained by subtracting each time-frequency point under the target-standard condition from the corresponding point under the target-target condition. Similarly, the corrected standard-target time-frequency map was acquired by subtracting the standard-standard condition from the standard-target time-frequency map. Therefore, the corrected analysis only compares the pure visual target-elicited alpha event-related desynchronization between the two conditions.

desynchronization observed under the target-target condition could simply result from addition of preceding auditory alpha ERD to the following visual alpha ERD and not from over-activation of the same visual networks. We addressed this issue by comparing corrected target-target and standard-target conditions in a supplementary analysis. Corrected conditions were created by subtracting the preceding auditory alpha ERD and the ERD elicited to visual standard stimuli from each condition (see legend for Fig 8). The comparison of pure target-elicited visual alpha ERD values between the two conditions showed that over-desynchronization of alpha under the target-target condition remained significant, albeit with prolongation of its latency (990–1530 ms). Therefore, whereas the early portion of observed over-desynchronization under the target-target condition was, to some extent, the result of contributions from auditory processing networks, the late over-desynchronization of alpha indicated over-activation of the same visual networks. This finding demonstrates for the first time that when the subroutine that is indexed by the visual P300 [11] is inhibited, the brain reacts to this absence by over-activation of the primary visual areas.

In this study, we found the precuneus, posterior cingulate cortex, and cingulate cortex to be generators of the visual P300. However, for alpha ERD, we found that the LORETA solution specified a large number of voxels as sources of visual alpha ERD. Nevertheless, the first 10 percent of voxels with the highest activation values included occipital and occipito-parietal areas such as the medial occipital gyrus and cuneus as major sources of visual alpha ERD. These results are in general agreement with those of previous studies for both P300 [49, 73, 74] and alpha ERD [49, 65, 66] and were used for subsequent connectivity analysis.

To track changes in the flow of information from the primary visual cortex to higher processing regions, we based our analysis on the Granger-based causal relationship between alpha ERD and the P300 [49], where it was demonstrated that the stream of information follows a one-way direction from visual alpha ERD sources to visual P300 sources. It is necessary to clarify that Granger-based causality results do not necessarily imply cause-and-effect interactions; rather, they point to a directed connectivity between two variables in that how well variance in

one variable can be predicted from variance in another variable earlier in time [56, 75]. Therefore, the Granger causal relationship observed between alpha ERD and P300 sources in [49] merely indicates that variance in waves extracted from P300 sources were better predicted when waves extracted from alpha ERD sources of earlier time were taken into account and not vice versa. Consequently, in this study, we sought to assess how the inhibition that is indexed by auditory P300 would affect direction of flow (prediction) of information (variance) between visual alpha ERD sources and visual P300 sources by comparing target-target and standard-target conditions.

Our effective connectivity comparisons did not provide any evidence of significant inhibition in the flow of information from visual alpha ERD sources to visual P300 sources. Thus, we could not find evidence for general widespread cortical inhibition at the time of the P300 time course, as suggested previously [12, 18, 22, 76]. Instead, the proposed inhibitory function that is indexed by P300 should be enacted at a higher processing stage (probably WM-related processes [48]) and should affect the ensemble of neurons around the loci of its generation (see the tertiary zones in [14]).

However, in connectivity analysis, the reverse flow of information from visual P300 sources to visual alpha ERD sources showed a sustained significant power reduction around the alpha band frequency. This reduction in power flow of alpha occurred in a narrow frequency band at 280 ms after the auditory target but rapidly expanded to a wide alpha band frequency after onset of the visual target and ended (1104 ms) at relatively the same time that alpha over-desynchronization in posterior regions started to occur (990–1530 ms; Fig 8). This trend reveals the following: at the same time as the auditory P300 was being generated, desynchronization of alpha, and subsequently when the visual P300 was inhibited, over-desynchronization of alpha was first driven from higher visual processing areas (at least to the scope of analysis in this research at the precuneus, posterior cingulate cortex, and cingulate cortex) to primary visual sensory areas (medial occipital gyrus and cuneus) and later propagated and recorded by electrodes in the occipital and parieto-occipital regions. This shows that when generation of the visual P300 in higher cortical areas is inhibited in a top-down manner, the brain actively struggles to increase the signal-to-noise ratio of afferent visual information by over-activating primary visual areas through reduction of alpha power. Based on Elbert and Rockstroh's model [77], we propose that when a cognitively important stimulus (target) enters the processing system, the brain responds to this stimulus in two ways: (1) by reducing the background noise in the EEG (ongoing EEG activity) and (2) by suppressing networks that are irrelevant to the processing of the target. The first response is deemed to be alpha ERD and the second response to be the P300, which results from the cumulative sum of positive charges in the extracellular spaces between apical dendrites of pyramidal cells where a synchronous action potential has not occurred [15]. Adopting this model, we can interpret our findings such that because of the inhibitory effect of the auditory P300, the brain had not provided the necessary suppression, that is, the visual P300, for processing of the upcoming visual target. Therefore, by compensatory over-activation of its second target-processing mechanism, that is, alpha ERD, the brain attempts to further diminish the background noise in the EEG so that target-relevant signals are enhanced. For an intuitive understanding of how alpha desynchronization can enhance the signal-to-noise ratio, Mathewson et al. [37] compared the brain with a football stadium full of people. In this simile, increased alpha activation is equivalent to the times when large crowds synchronously cheer their teams. During these high-amplitude synchronized cheers, individual shouts (afferent sensory signals) have a lower probability of being heard. However, if the synchronous applause becomes desynchronized and scatters over time (alpha ERD), there is an increased probability of a loud shout being heard. Based on our findings, we extend this simile using Elbert & Rockstroh's model [15, 77] by proposing that there are

officers (higher brain regions responsible for monitoring incoming data) in the stadium who shut down (P300) irrelevant shouts when a VIP person (target) starts to speak. Whenever these officers are incapable of silencing the stadium (suppression of P300), and thus the chances of irrelevant shouts increases (extraneous information), for the shout of the VIP person to be heard, the synchronous cheering of the crowd needs to be further desynchronized (over-desynchronization of alpha in sensory areas). This model can also help to explain behavioral findings with regard to P300 inhibition. While over-desynchronization of alpha in sensory regions increases the signal-to-noise ratio of incoming relevant information (target), since this mode of activation (in the absence of inhibition by P300) is suboptimal, the time taken to react to the target increases [18, 23, 67]. Moreover, when the incoming signal itself is weak, such as near-threshold stimuli in signal detection tasks, there is a decrease in the sensitivity parameter, which is an index of the ability of the perceptual system to distinguish between signal and noise [19, 20, 78].

At this point, it is necessary to assert that inhibition as a fundamental and physiological function of the P300 in the brain is not essentially at odds with prominent theories regarding it cognitive basis [18]. For example, inhibition as the basic function of P300 can be considered to be a limited resource [6, 79–81], the capacity of which is consumed by cognitively important information (targets). More importantly, linking the inhibitory function of P300 to WM (context) updating theory [11] can provide new insights into the mechanisms of WM in the brain. The WM updating theory of P300 [10, 82] assumes that the brain consistently stores a model (context) from the environment in WM. Whenever a significant change in the stream of incoming information with regard to the environment occurs, the brain updates the pre-existing model in WM and this update is represented by the amplitude of the P300. The P300 as an indicator of the cognitive WM updating process can thus be a result of suppression of extraneous neural assemblies at the time of updating [15, 48]. Moreover, given that our findings in this study have shed more light on the scope and stage where inhibition is enacted, we can infer that inhibition by the P300 is restricted to the network of WM representations at the loci of its generation, that is, temporal or parietal structures related to memory storage [48]. Linking inhibition to WM updating theory can also provide a cognitive framework for explaining the behavior of alpha observed in this study. Klimesch et al. [41] proposed that alpha event-related synchronization within Baddeley's concept of WM [83] is a representation of top-down inhibition that is at the service of central executive functions. Top-down processes are attentional-based control functions that keep cognitive processes focused on specific aspects of tasks. This control is essentially implemented through inhibition of task-unrelated cortical areas to prevent interference from other cognitive processing systems [41]. By relating this hypothesis to Elbert and Rockstroh's model [15, 77], we assume that executive functions of the WM not only apply top-down inhibition to task-irrelevant cortical areas but also selectively activate task-related brain regions. In their original hypothesis, Klimesch et al. [41] stated that alpha ERD is only observed in bottom-up processes. However, in the present study, our effective connectivity analysis clearly pointed to top-down activation of primary visual sensory areas through modulation of alpha ERD that was implemented by higher visual processing regions. Therefore, based on this research, we argue that the executive functions of WM behave as follows: by (1) inhibiting irrelevant neural networks in higher visual processing areas (parietal or temporal association regions), which are probably loci of representations of the visuospatial sketchpad [84], they update WM representations in the visuospatial sketchpad (P300) and (2) by reducing alpha power in primary visual areas, they increase the signal-to-noise ratio of incoming information, which is later used for WM updating processes provided that they are identified to be cognitively important. Hence, based on this framework, the over-desynchronization of alpha ERD, at the time when the P300 is suppressed, can result from an

attempt by the executive functions to compensate for malfunction of inhibition in the visuo-spatial sketchpad (suppression of P300) by further reducing EEG's background noise. However, this compensatory mechanism is suboptimal and results in delayed reaction times and poor sensitivity measures.

## Limitations and further suggestions

In this research design, due to the nature of our aim, we could not shuffle the positions of auditory and visual stimuli. Therefore, our findings are limited to the effect of the auditory oddballs on visual processes, meaning that they cannot be generalized to interactions of P300 with other modalities, such as the effect of the visual oddballs on auditory processing and/or somatosensory P300s. Moreover, our source localization, as well as effective connectivity analysis, are bound to limitations in EEG methodology, such as volume conduction. This could cast doubt on the accuracy of our findings. Therefore, we suggest that future studies aim to replicate our findings using other methodologies, such as magnetoencephalography or intracranial EEG. Nevertheless, disentangling two mechanisms of WM in processing target stimuli has implications for clinical and cognitive research. For example, future studies could assess the extent to which the excitatory (alpha ERD) or inhibitory (P300) sub-mechanism of WM is impaired in patients with attention deficit hyperactivity disorder. Moreover, by presenting transcranial magnetic stimulation (TMS) pulses during the time course corresponding to the peak of the P300, researchers can further assess whether TMS-evoked potentials are inhibited in comparison with conditions where these pulses are presented outside the peak time of the P300. Finally, our findings have implications for brain-computer interface research. Given that generation of distinct consecutive P300 waves with short stimulus onset asynchrony of at least 400 ms seems unlikely, we propose over-desynchronization of alpha as an alternative index for detection of rapid brain signals in response to paired target sets.

## Acknowledgments

We would like to acknowledge the Iranian National Brain Mapping Laboratory (NBML), Tehran, Iran, for providing data acquisition service for this research work.

## Author Contributions

**Conceptualization:** Amirmahmoud Houshmand Chatroudi, Reza Rostami.

**Data curation:** Amirmahmoud Houshmand Chatroudi, Ali Motie Nasrabadi.

**Formal analysis:** Amirmahmoud Houshmand Chatroudi, Ali Motie Nasrabadi.

**Funding acquisition:** Yuko Yotsumoto.

**Investigation:** Amirmahmoud Houshmand Chatroudi, Reza Rostami.

**Methodology:** Amirmahmoud Houshmand Chatroudi, Ali Motie Nasrabadi.

**Project administration:** Amirmahmoud Houshmand Chatroudi, Reza Rostami.

**Resources:** Amirmahmoud Houshmand Chatroudi, Reza Rostami, Yuko Yotsumoto.

**Software:** Yuko Yotsumoto.

**Supervision:** Reza Rostami, Ali Motie Nasrabadi, Yuko Yotsumoto.

**Validation:** Reza Rostami, Ali Motie Nasrabadi, Yuko Yotsumoto.

**Visualization:** Amirmahmoud Houshmand Chatroudi.

Writing – **original draft:** Amirmahmoud Houshmand Chatroudi.

Writing – **review & editing:** Reza Rostami, Yuko Yotsumoto.

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
