## [Decision Letter · Decision Letter 0]

15 Jan 2021

PONE-D-20-40155

Inhibitory Effect of the Auditory P300 on Transmission of Visual Sensory Information

PLOS ONE

Dear Dr. Rostami,

Thank you for submitting your manuscript to PLOS ONE. After careful consideration, we feel that it has merit but does not fully meet PLOS ONE’s publication criteria as it currently stands. Therefore, we invite you to submit a revised version of the manuscript that addresses the points raised during the review process.

This is an interesting and well-conducted study. As the authors can see both reviewers expressed positive comments on this manuscript, although they also raised some minor points that should be addressed. Thus, I suggest the authors address the suggested changes and resubmit the manuscript as soon as they can. Thank you.

We look forward to receiving your revised manuscript.

Kind regards,

Vilfredo De Pascalis

Academic Editor

PLOS ONE

2. Please improve statistical reporting and refer to p-values as "p<.001" instead of "p=.000". Our statistical reporting guidelines are available at https://journals.plos.org/plosone/s/submission-guidelines#loc-statistical-reporting.

Additional Editor Comments:

This is an interesting and well-conducted study. As the authors can see both reviewers expressed positive comments on this manuscript, although they also raised some minor points that should be addressed.

Thus, I suggest the authors address the suggested changes and resubmit the manuscript as soon as they can.

Thank you.

Reviewers' comments:

Reviewer's Responses to Questions

**Comments to the Author**

1. Is the manuscript technically sound, and do the data support the conclusions?

Reviewer #1: Yes

Reviewer #2: Yes

2. Has the statistical analysis been performed appropriately and rigorously? 

Reviewer #1: Yes

Reviewer #2: Yes

3. Have the authors made all data underlying the findings in their manuscript fully available?

Reviewer #1: No

Reviewer #2: Yes

4. Is the manuscript presented in an intelligible fashion and written in standard English?

Reviewer #1: Yes

Reviewer #2: Yes

5. Review Comments to the Author

Reviewer #1: Please see the attached. Please see the attached. Please see the attached. Please see the attached. Please see the attached. Please see the attached. Please see the attached. Please see the attached. Please see the attached.

Reviewer #2: Review of PONE-D-20-40155

In this article the authors explore the relation between auditory and visual oddball stimuli by presenting an auditory tone followed by a visual square. The auditory stimulus can be an oddball or a standard, and the visual stimulus can be an oddball or a standard. This gives 4 types of trials and allows the investigation of the EEG waveform when either, or both stimuli are oddballs, and their interactions.

In my view the most interesting finding comes from Panel D of Figure 2, which shows a lack of P300 to the visual oddball when it is preceded by an auditory oddball, which I assume is a replication of Nash and Fernandez. The present study seems to extend that research by looking at event-related desynchronization (ERD; essentially alpha power) and changes in functional connectivity.

The data is competently collected, analyzed, and processed into higher-level analyses using standard and appropriate techniques.

A novel finding is with respect to the P100 of the visual stimulus. The authors make this claim: “our exploratory ERP analysis showed that when following auditory targets, the P100 evoked in response to standard and target visual stimuli is not inhibited; rather, it is encouraged.” It looks like this is happening because the visual P100 is riding on top of the auditory P300 from the target, and that is causing the P100 differences. If this is correct, the authors should discuss this.

Beyond this, I feel like the authors have done a nice job addressing their hypotheses with respect to connectivity and the relation between the P300 and alpha. I see no other major flaws.

I sign all reviews,

Tom Busey

6. PLOS authors have the option to publish the peer review history of their article (what does this mean?). If published, this will include your full peer review and any attached files.

Reviewer #1: No

Reviewer #2: **Yes: **Thomas Busey

---

## [Author Response · Author response to Decision Letter 0]

5 Feb 2021

We wish to express our appreciation to the editor and the reviewers for their insightful comments, which have helped us significantly improve our manuscript. We hope that the revisions improve the paper such that the editor and the reviewers now deem it worthy of publication in the PLOS ONE journal. 

Next, we offer detailed responses to the editor and reviewers’ comments. The editor and reviewers’ comments are shown in BOLD font, our responses are in blue, and the sentences in the revised manuscript are shown in italic.

Editor

We appreciate you bringing this issue to our attention. We double checked the manuscript with PLOS ONE's style requirements and made amendments accordingly: we revised the manuscript's first page which includes author's names and affiliations according to the sample provided. Moreover, we noticed some mistakes in Table 1 and Fig 7's titles and label style which we corrected. Finally, we ensured we have met other style requirements such as heading levels and file naming standards.

2. Please improve statistical reporting and refer to p-values as "p<.001" instead of "p=.000". Our statistical reporting guidelines are available at https://journals.plos.org/plosone/s/submission-guidelines#loc-statistical-reporting.

Thank you very much for catching this problem. We have checked and revised p-values according to PLOS ONE's statistical reporting guidelines. Thus, all p-values less than p < .001 are expressed as p < 0.001.

We apologize for not noticing this issue beforehand. Upon your notification, the correspondent author has linked his ORCID iD to the PLOS ONE author information.

REVIEWER 1

On each trial, an auditory odd-ball (or a standard) was presented 400ms prior to a visual odd-­ball (or a standard). Participants counted the number of auditory odd-­balls and quickly responded to each visual odd-­ball. The auditory odd­balls generated the P300 ERPs at 300-500ms (all times are measured relative to the initial auditory onset), and slowed the subsequent responses to the visual odd-­balls. The P300 ERPs to a visual odd-­ball (650-­900ms) was suppressed when an auditory odd-­ball preceded the visual odd-ball. An auditory odd-ball reduced posterior alpha power at 300-500ms (prior to the onset of visual odd-ball at 400ms) and a visual odd-­ball (when preceded by an auditory odd-­ball) reduced posterior alpha power at 900-1300ms. In addition, the alpha power at 300-1000ms in the LORETA­defined posterior alpha sources was significantly correlated with the preceding P300­EEG (up to 500ms) in the LORETA-defined P300 sources (with no delayed correlations in the reversed direction). These are my understanding of the primary results. They appear to be reliable (given the statistics provided) and obtained with appropriate methods, thus are likely to contribute to the relevant literature. I personally think that the theoretical discussions are excessive given the scope of the results, but the authors should have the freedom to discuss them so long as speculative inferences are clearly indicated as such. Below are some comments and suggestions that the authors may be interested in considering in their revision.

We would like to express our sincere appreciation to you for the meticulous reading of our paper and insightful and critical points you have made. Thanks to your comments, we revised our manuscript and have clarified obscure sections, sentences and phrases. Below are the comments you kindly gave us and the explanation to how these were implemented into the paper.

1. The authors talk about “the inhibitory effect of the P300,” but P300 is a phenomenologically defined ERP component. Any inhibitory effect must be exerted by some underlying processes (e.g., auditory odd-­ball detection, the updating of odd-­ball counts in WM, etc.) that generate P300 recorded from the scalp.

Thank you very much for raising this important point. Indeed, we fully agree with your opinion. We miswrote the phrase "the inhibitory effect of the P300". Accordingly, we have changed the title of the paper to indicate P300 is an index of inhibition and not the cause of inhibition. 

"Effect of Inhibition Indexed by Auditory P300 on Transmission of Visual Sensory Information"

Moreover, we re-read the whole manuscript and replaced all phrases that implied P300 is the cause of inhibition with phrases that clearly state P300 is an index of inhibition. Finally, we have incorporated your comment into the first paragraph of the Discussion section (lines 653-657) in order to clarify this point:

"It is important to note that any inhibitory effect associated with P300 is in fact driven by a subroutine that is summoned based on task demands. Thus, in this study, the inhibitory effect that is indexed by the auditory P300 is exerted by the underlying mechanisms that are required for oddball target detection and/or the updating of oddball counts in WM."

2. The reference [25 (191)] for the neural sources of alpha oscillations may be a bit dated. A study by Bollimunta et al. may be relevant (Bollimunta, Chen, Schroeder, and Ding, 2008, J Neurosci). Also, equating posterior alpha to sensory inhibition may be a bit simplistic. Posterior alpha oscillations have also been linked to oscillations of visual awareness (e.g., Mathewson and colleagues), the sequential structuring of incoming visual information (e.g., Jensen et al., 2014, Trends in Neurosciences), etc.

Thank you for bringing this point to our attention. 

Neural basis of alpha: since the neural bases of alpha are subject of intensive debate and this paper does not deal with them, we decided to remove the paragraph that was devoted to neural basis of alpha in the brain. Instead, based on your suggestion, we updated our references and briefly mentioned potential neural sources of alpha for the interested readers, including the paper you kindly introduced (lines 117-119). 

"Alpha waves are 8–13-Hz oscillations that debatably originate from corticocortical (e.g., layer 5 in V2, V4 and layer 2/3 in inferotemporal cortex [24]; layer 4 in V1 [25]; higher-order visual and somatosensory areas, [26]) or thalamocortical connections [27–30]… "

Functional meaning of alpha: we believe the current literature regarding functional meaning of alpha converges on the idea that alpha rhythm indicates inhibition. Periodicity of the inhibition that is signaled by alpha rhythm is in fact the reason why alpha power has been linked to fluctuations of awareness (as the pulsed-inhibition theory of alpha put forward by Mathewson and colleagues suggests [1,2]) or temporal coding of visual processing (as Jensen et al [3] rely on rhythmic inhibition as the primary function of alpha). Consequently, thanks to your suggestion, we revised the section that dealt with functional meaning of alpha to clarify this issue (lines 119-126).

"An alpha rhythm is classically deemed to be an 'idling rhythm' [31] or an index of cortical inactivation that signals a closed thalamic gate [32]. More recent studies, have associated increases in alpha power (known as event-related synchronization, ERS) with active inhibition (e.g., attention-modulated suppression of sensory input [33–36]; pulsed-inhibition of ongoing neural activity [37,38]; periodic inhibition of task irrelevant brain areas [39–41]). On the other hand, reduction of alpha power (known as event-related desynchronization, ERD) has been associated with an open thalamic gate [32] and is considered to be an index of cortical excitation or release of inhibition [41,42]."

3. Lines 132 to 135. The two statements seem to be inconsistent. The first states that P300 predicts alpha ERD. The second, which is supposed to confirm the directionality of the first statement, states that information flows from the alpha ERD sources to the P300 sources.

Thank you very much for kindly pointing out this unclear part. We revised the section to eliminate potential misunderstandings (lines 130-135). 

"By using multiple regression analysis, a significant relationship was found between the P300 and alpha ERD, such that the peak and latency of alpha ERD could be predicted by the peak and latency of P300 [47]. However, this initial finding was challenged [48] and in a recent study [49] using effective connectivity, based on Granger causality, it was recognized that there is a consistent flow of information from cortical alpha ERD sources to the sources that generate the P300 and not vice versa."

4. Line 272. What is the “length” of a Morlet wavelet? It is Gaussian-windowed so would its range not be infinite? The authors might provide the temporal standard deviation instead.

We are thankful that you raised this point. We revised the section to eliminate potential misunderstandings (lines 274-285). All parameter settings for complex Morlet wavelet decomposition were set according to recommendations specified in [4]; chapters 12 and 13.

"…using complex Morlet wavelet decomposition on single-trial EEG data with custom MATLAB scripts according to the following formula [56]:

CMW=e^(〖-t〗^2⁄〖2s〗^2 ) e^i2πft

The length of the Morlet wavelet (t) was set to 4 s (range +2 to -2 centering on 0 s based on recommendations in [56]). The central frequency parameter (f) was adjusted to the range of 1–30 Hz in 0.5-Hz increments. The standard deviation or the width of the Gaussian (s) was obtained using following formula: 

s= n/2πf

The number of wavelet cycles (n) was set to the logarithmically spaced range of 4–10 cycles so that it increased proportionately with central frequency (thus, the range of standard deviation (s) was 0.63–0.05). An increase in the number of cycles as a function of central frequency was implemented to balance the precision of time and frequency [56]. "

5. The permutation-based statistical testing described in lines 308+ uses the 99.5th percentile criterion whereas that described in lines 397+ uses the 95th percentile criterion. Is one of them a typo, or was there a rationale behind using different criteria? 

Thank you very much for kindly pointing out this unclear part. We clarified our choice of p-values and multiple comparison correction methods separately for alpha ERD analysis (lines 329-334) and effective connectivity analysis (lines 429-434).

Lines 329-334: "In comparison to cluster-based correction (see effective connectivity subsection), pixel-based multiple comparison correction, is considered to be a more stringent correction method for time-frequency maps [56]. Since the exploratory analysis was performed post hoc to the hypothesis-driven analysis, we used this method together with a strict p-value (p < 0.01) to assure our results have sufficient statistical rigor."

Lines 429-434: "In comparison to pixel-based correction, cluster-based method is a less stringent multiple comparison correction. We used this method together with a less strict p-value (p < 0.05) to allow for detection of more areas of difference between two conditions across whole frequency and time point space separately for flow of information from visual alpha ERD sources to visual P300 sources and vice versa."

Moreover, we clarified that we used two-tailed p-values for alpha ERD and effective connectivity analysis respectively on lines 322 and 420.

Additional explanation: In pixel-based multiple comparison correction, at each iteration during permutation testing, the largest maximum and largest minimum values are stored, generating two distributions of extreme vales. Subsequently, for two-tailed p-value < 0.01, the values that correspond to the 99.5th and 0.5th percentiles are taken as thresholds which are then applied to grand average time-frequency map differences [4]; pages 470-471. In cluster-based multiple comparison correction, after obtaining the null distribution, the time-frequency maps are normalized to z-values and are thresholded (two-tailed p < 0.05 in our effective connectivity analysis; this criteria is called precluster threshold). After thresholding, cluster of pixels with largest values from each sample of null distribution are extracted and stored, generating a histogram of cluster sizes expected under the null hypothesis. The value at 95 percentile of this distribution (corresponding to p < 0.05) is taken as the cluster correction threshold. Subsequently, this threshold is used to remove clusters from the precluster-thresholded grand average time-frequency map differences [4]; pages 471-475. 

We revised the cluster-based multiple comparison correction section by adding labels "precluster threshold" and "cluster correction threshold" to avoid confusion (lines 423-426). 

"…When using this method, the distribution of the largest adjacent points that form significant time-frequency clusters at p < 0.05 (two-tailed; precluster threshold) under the null hypothesis are extracted from the permuted data. After obtaining this distribution, the cluster size value, which corresponds to the 95th percentile of the distribution, is acquired (cluster correction threshold; p < 0.05)…"

Finally, on lines 328 and 428, respectively, we encouraged the interested reader to see a more detailed explanation of these methods at [4].

6. In Figure 3, it is unclear to me what ROI is indicated by the red square.

Thank you for pointing it out. We revised Fig 3 caption to eliminate potential misunderstandings.

"...(D) Comparison of power in the time-frequency ROI between the target-target and standard-target conditions. The red square indicates the ROI where the effect of the inhibition that is indexed by the auditory P300 was expected to be observed on the time-frequency map of difference between target-target and standard-target conditions (8–13 Hz, 616–822 ms; hypothesis driven analysis, see Method section). This ROI is also marked on panels A and B with red square to indicate its relative position separately on baseline-corrected target-target and standard-target time-frequency maps…"

7. Additional discussions on the Granger analysis may be helpful. First, the authors should NOT equate significant Granger values with causality. It is just a correlation analysis with a series of temporal delays (10ms increment in the study) within a time window (500ms in the study). For instance, a significant Granger value from P300 to alpha ERD just means that up to 500ms prior P300 EEG (with appropriate weighting of different prior time points) predicts the current alpha ERD. Significant Granger values are consistent with the interpretation that P300 is directly causing alpha ERD. However, it is also consistent with an alternative interpretation that some processes related to auditory odd-­ball detection generate BOTH P300 and alpha ERD, but the latter with a longer delay, which could make Granger values from P300 to alpha ERD significant. That is, significant Granger values are consistent with causality/connectivity, but they do NOT imply causality/connectivity. I think that it is very important to respect this distinction.

Thank you very much for raising this very important point. We incorporated your comment into the discussion section right before discussing the effective connectivity results on lines 751-761.

"It is necessary to clarify that Granger-based causality results do not necessarily imply cause-and-effect interactions; rather, they point to a directed connectivity between two variables in that how well variance in one variable can be predicted from variance in another variable earlier in time [56, 75]. Therefore, the Granger causal relationship observed between alpha ERD and P300 sources in [49] merely indicates that variance in waves extracted from P300 sources were better predicted when waves extracted from alpha ERD sources of earlier time were taken into account and not vice versa. Consequently, in this study, we sought to assess how the inhibition that is indexed by auditory P300 would affect direction of flow (prediction) of information (variance) between visual alpha ERD sources and visual P300 sources by comparing target-target and standard-target conditions." 

Moreover, we re-read the manuscript and added the phrase "one-way direction" in parenthesis whenever we used the term "causal relationship" to clarify that the intended meaning of causality in this paper is not equivalent to cause-and-effect interactions.

8. To make the case that the Granger effect from P300 to ERD is uniquely in the alpha range, it might be useful to show the frequency dependence of the Granger value (to show that it peaks at alpha).

Thank you very much for your insightful suggestion. We added subplots to the time-frequency maps of Fig 7. In these subplots the spectral power density over the whole epoch, ranging from time = 0 s up to the end of the epoch, is averaged and is plotted versus frequency axis separately for both visual P300 to visual ERD sources and visual ERD to visual P300 sources. Accordingly, we added proper descriptions for these subplots at Fig 7 caption.

Part of Fig 7 caption that describes the newly added subplots: "The subplots on the right side of time-frequency maps show spectral power density differences between two conditions (corresponding to the time-frequency maps on their left) that are averaged across the whole epoch (starting from time = 0 ms up to the end of the epoch). The spectral power density subplot of flow of information from visual P300 to visual alpha ERD sources reveals a major power reduction difference between two conditions that peaks at 9 Hz." 

9. I do not understand how the time-frequency power (dB) plots shown in Figure 7 are related to the Granger values. Are the power values weighted by a Granger index (e.g., the logarithm of the ratio of the univariate error variance to the bivariate error variance)? Some details would be appreciated.

We appreciate you for pointing this out to us. We clarified this by adding details on lines 403-407 in the method section. 

"After fitting separate vector autoregressive (VAR) models to a sequence of highly-overlapping time windows, by Fourier transforming the fitted time-varying VAR coefficients, SIFT extracts complex spectral density [61]. The spectral power density is then reported in dB units, 10*log10 of complex spectral density."

Moreover, for the Fig 7, we clarified what dB units represent by adding the description at the end of the Fig 7 caption.

"… dB, unit of spectral power density (see Method section) "

10. The asymmetry of Granger results between ERD -­> P300 and P300 -­> ERD is emphasized. However, is it not the case that Granger predictions could not go in the direction of ERD to P300 by design? If I understood the study correctly, the auditory stimuli always preceded the visual stimuli and the auditory-evoked P300 peaked prior to the visual onset. That is, the auditory-evoked P300 preceded the visually-induced posterior alpha ERD. Then, how could the latter causally influence the former? A clarification would be appreciated.

Thank you very much for bringing this point to our attention. 

We would like to clarify that the purpose of using Granger-based effective connectivity in this study was to assess how the inhibition which is signaled by auditory P300 would affect flow of information from visual alpha ERD sources to visual P300 sources (lines 144-147, 360-363, 611-612, 652-653, and 748-751). The rationale behind this investigation came from a study by Peng, et al. [5] where they showed direction of granger prediction is from alpha ERD sources to P300 sources (lines 133-135, 652-653, and 749-751). Accordingly, we extracted single-trial brain waves (lines 364+) from estimated visual P300 sources (lines 337+) and visual ERD sources (lines 351+). Next, we assessed how spectral power density of Ganger-based effective connectivity from visual alpha ERD sources to visual P300 sources and vice versa would be different between two main conditions of our study; standard-target and target-target conditions (360+). The rationale behind comparison of these two conditions was based on the fact that when visual oddball target is presented at around the peak of the preceding auditory P300, formation of the visual P300 is suppressed (lines, 95-98 and Fig 2). Our effective connectivity results that are showcased in Fig 7 indicate that, in comparison of the aforementioned conditions, there is no significant power change in spectral density of flow of information from visual alpha ERD sources to visual P300 sources; meaning we observed no effect of the inhibition (indexed by auditory P300) on transmission of information in that direction (discussed on lines 762-768). However, surprisingly, analysis of the reverse direction showed a significant difference in spectral power density between the aforementioned conditions (discussed on lines 769+).

In light of this brief review of the current study, we would like to ascertain that we do not make any claims about whether auditory P300 can Granger predict (causally influence) visual alpha ERD. We would like to emphasize that we restricted our source localization analysis to visual P300 and visual ERD sources and subsequently, we assessed how inhibition indexed by auditory P300 would potentially affect flow of visual information between these two sources by assessing spectral power density differences between standard-target and target-target conditions. Thus, we did not examine effective connectivity between estimated P300 auditory sources with visual alpha ERD sources. Moreover, to our knowledge, the Granger causal relationship from alpha ERD to P300 sources has only been observed in same modalities [5] and no study has so far explored how waves extracted from sources of P300 from one modality would Granger predict waves from sources of alpha ERD in another modality. In fact, this idea seems to be a very compelling direction for future studies. Finally, as we already have mentioned in the 'limitations and further suggestions' section of the manuscript, we would like to reiterate that findings observed in this study are strongly limited to the effects that processing of auditory oddballs would have on visual processes. Thus, we hope future studies, by counterbalancing or replacing position of auditory and visual oddballs, shed more light on generalizability of our findings. 

We re-read the manuscript and wherever P300 or alpha ERD sources were under discussion, we added the term "visual" before P300 and alpha ERD sources to assure there will be no confusion with that regard. We also implemented the same revision to top row titles inside Fig 7 for more clarification. 

REVIEWER 2

In this article, the authors explore the relation between auditory and visual oddball stimuli by presenting an auditory tone followed by a visual square. The auditory stimulus can be an oddball or a standard, and the visual stimulus can be an oddball or a standard. This gives 4 types of trials and allows the investigation of the EEG waveform when either, or both stimuli are oddballs, and their interactions.

In my view, the most interesting finding comes from Panel D of Figure 2, which shows a lack of P300 to the visual oddball when it is preceded by an auditory oddball, which I assume is a replication of Nash and Fernandez. The present study seems to extend that research by looking at event-related desynchronization (ERD; essentially alpha power) and changes in functional connectivity.

The data is competently collected, analyzed, and processed into higher-level analyses using standard and appropriate techniques.

Thank you very much for your diligent reading and your accurate understanding of our study. 

A novel finding is with respect to the P100 of the visual stimulus. The authors make this claim: “our exploratory ERP analysis showed that when following auditory targets, the P100 evoked in response to standard and target visual stimuli is not inhibited; rather, it is encouraged.” It looks like this is happening because the visual P100 is riding on top of the auditory P300 from the target, and that is causing the P100 differences. If this is correct, the authors should discuss this.

We appreciate you for pointing this important issue out. Accordingly, we have edited the manuscript (lines 672-676) to draw a more conservative conclusion with regard to the observed differences for the visual P100. Moreover, we replaced the term "enhancement" with "intactness" on lines 689 and 690 to reflect this idea.

"…; rather, it is encouraged. However, this amplitude enhancement should be interpreted with caution as the visual P100 is added on the top of the preceding auditory P300. Thus, we conservatively interpret this enhancement as a mere indication that the inhibition that is indexed by P300 has not suppressed the formation of the subsequent visual P100. "

Beyond this, I feel like the authors have done a nice job addressing their hypotheses with respect to connectivity and the relation between the P300 and alpha. I see no other major flaws.

Thank you very much for your positive feedback on our study. We sincerely appreciate your meticulous reading and your precise understanding of the manuscript.

References

1. Mathewson KE, Lleras A, Beck DM, Fabiani M, Ro T, Gratton G. Pulsed Out of Awareness: EEG Alpha Oscillations Represent a Pulsed-Inhibition of Ongoing Cortical Processing. Front Psychol. 2011;2: 1–15. doi:10.3389/fpsyg.2011.00099

2. Mathewson KE, Gratton G, Fabiani M, Beck DM, Ro T. To see or not to see: Prestimulus α phase predicts visual awareness. J Neurosci. 2009;29: 2725–2732. doi:10.1523/JNEUROSCI.3963-08.2009

3. Jensen O, Gips B, Bergmann TO, Bonnefond M. Temporal coding organized by coupled alpha and gamma oscillations prioritize visual processing. Trends Neurosci. 2014;37: 357–369. doi:10.1016/j.tins.2014.04.001

4. Cohen MX. Analyzing Neural Time Series Data: Theory and Practice. Cambridge: MIT Press; 2014. 

5. Peng W, Hu L, Zhang Z, Hu Y. Causality in the Association between P300 and Alpha Event-Related Desynchronization. Maurits NM, editor. PLoS One. 2012;7: e34163. doi:10.1371/journal.pone.0034163

---

## [Editor Report · Decision Letter 1]

8 Feb 2021

Effect of Inhibition Indexed by Auditory P300 on Transmission of Visual Sensory Information

PONE-D-20-40155R1

Dear Dr. Rostami,

We’re pleased to inform you that your manuscript has been judged scientifically suitable for publication and will be formally accepted for publication once it meets all outstanding technical requirements.

Kind regards,

Vilfredo De Pascalis

Academic Editor

PLOS ONE

Additional Editor Comments (optional):

I want to compliment the authors for the excellent quality of their work. I see that all the points raised by the reviewers have been addressed, making a great improvement to the current revised paper.

I have really appreciated this paper for the highly sophisticated signal processing methods used. Thus, I reached the conclusion that the current version of the manuscript can be accepted for publication.
---

## [Editor Report · Acceptance letter]

11 Feb 2021

PONE-D-20-40155R1 

Effect of Inhibition Indexed by Auditory P300 on Transmission of Visual Sensory Information 

Dear Dr. Rostami:

I'm pleased to inform you that your manuscript has been deemed suitable for publication in PLOS ONE. Congratulations! Your manuscript is now with our production department. 

Kind regards, 

on behalf of

Prof. Vilfredo De Pascalis 

Academic Editor

PLOS ONE